# MULTI-SCALE MEMORY FUSION WITH DYNAMIC DECAY FOR COHERENT TEXT-TO-MOTION GENERATION

## ABSTRACT

Text-to-3D human motion generation has emerged as a critical challenge in human-AI interaction, with transformative applications spanning virtual reality, robotic control and digital content creation. While recent advances in diffusion models and transformer architectures have significantly improved motion quality, we identify two fundamental limitations that persist in state-of-the-art methods: (1) suboptimal utilization of multi-scale historical context leading to motion discontinuity, and (2) uniform temporal weighting that fails to capture phase-dependent feature importance in complex motion sequences. To address these challenges, we propose FADM (Feedback-Augmented Decay Motion Model), a novel framework that introduces three key innovations: a hierarchical memory fusion module with learnable scale adapters for preserving both local kinematics and global action semantics, an exponentially decaying temporal attention mechanism grounded in human motion dynamics, and a semantic-consistent autoregressive feedback loop ensuring long-range coherence. Extensive experiments demonstrate our method's state-of-the-art performance, achieving a 22.2% FID reduction on HumanML3D, 64.7% improvement in Top-1 accuracy, and 30.9% better generalization on KIT-ML, while maintaining competitive motion diversity (Multimodality score: 1.283±0.044). Beyond its immediate applications, FADM establishes a new paradigm for temporal modeling that can potentially benefit various conditional generation tasks including video synthesis and robotic motion planning.

## 1 INTRODUCTION

The ability to translate natural language into coherent and lifelike 3D human motions is poised to become a cornerstone technology in the era of embodied AI. With the convergence of virtual reality (VR), the metaverse, gaming, and human-robot interaction, text-to-motion generation is rapidly shifting from a niche research problem to an essential enabler for next-generation digital experiences. Imagine describing "a dancer spins twice and leaps gracefully forward" and instantly seeing a virtual character perform it with natural continuity and semantic fidelity. This capability could redefine VR immersion, accelerate animation pipelines in film and game production, and enable more adaptive human-robot collaboration in unstructured environments.

As the field evolves, text-to-motion generation has witnessed substantial progress, yet it remains intrinsically challenging due to the need to capture both fine-grained local kinematics and long-range semantic consistency under natural language guidance. Early approaches, such as the variational autoencoder (VAE)-based T2M (Guo et al., 2022), attempted to learn probabilistic mappings between text and motion; however, their limited semantic expressiveness resulted in coarse and less faithful outputs. To overcome this limitation, autoregressive models like T2M-GPT (Zhang et al., 2023a) and diffusion-based methods, such as MotionDiffuse (Zhang et al., 2024a), have been introduced, leveraging Transformer architectures and discrete motion tokens. These approaches significantly improved motion fidelity and diversity. However, their reliance on uniform temporal modeling often causes inconsistencies in long sequences, as they struggle to balance short-term smoothness with long-term coherence.

More recent studies have explored hierarchical and component-based designs to further enhance motion generation. For example, MoMask (Guo et al., 2024) adopts a bidirectional masked mod-

eling framework with a hierarchical Transformer to strengthen contextual representations, while Mamba-based methods such as KMM (Zhang et al., 2024b) and Motion Mamba (Zhang et al., 2024d) leverage efficient state space models to capture temporal dependencies—KMM introduces keyframe attention to highlight critical motion anchors, and Motion Mamba achieves scalable modeling through hierarchical temporal scanning and bidirectional spatial reasoning. Component-level approaches also demonstrate unique benefits: ParCo (Zou et al., 2024) designs limb-specific generators to refine local details, and MoGenTS (Yuan et al., 2024) employs joint-level discretization to alleviate the information loss from global quantization.

However, despite these advancements, existing methods still face fundamental challenges. Specifically, current generation approaches often fail to fully utilize multi-scale historical context, resulting in a lack of temporal coherence and consistency in generated motions. Moreover, the uniform temporal weighting mechanism does not accurately reflect the varying importance of features at different stages of the motion sequence, which limits the model's flexibility and adaptability.

Particularly for the second point, we introduce a dynamic decay factor inspired by the memory decay patterns observed in cognitive science. When processing continuous motions, humans tend to rely more heavily on recent movements while their attention to distant historical information diminishes unevenly, exhibiting a nonlinear decay of memory weights. FADM simulates this natural phenomenon by designing an exponentially decaying temporal attention mechanism, enabling the model to allocate attention across time steps more reasonably and thus better capture the dynamic changes and critical information at different stages of motion sequences.

To address these issues, this paper proposes a novel framework named FADM (Feedback-Augmented Decay Motion Model), which effectively enhances the fusion of historical information and the modeling of temporal importance through multi-scale autoregressive feedback and dynamic decay weights. The main contributions are as follows:

- We propose a multi-scale memory fusion module with a dynamic feedback gating mechanism, which effectively integrates historical motions with current textual conditions, thereby enhancing contextual modeling and ensuring semantic consistency of generated sequences.

- We introduce a dynamic temporal weighting mechanism based on exponential decay, which adaptively allocates feature importance across time steps, suppressing excessive interference from early information and enhancing rhythm adaptability.

- We conduct experiments on two benchmark text-to-motion datasets, HumanML3D and KIT-ML. Results demonstrate that our method significantly reduces FID scores and improves multiple accuracy metrics, validating its effectiveness and superiority.

The rest of this paper is organized as follows: Section 2 reviews related work in text-to-motion generation; Section 3 details the proposed FADM framework and its key module designs; Section 4 presents and analyzes experimental results across multiple datasets; and Section 5 concludes the paper and discusses future research directions.

## 2 RELATED WORK

### 2.1 TEXT-TO-MOTION GENERATION

The development of text-to-motion generation has evolved from traditional approaches to deep learning-based methods. Early studies primarily employed deterministic techniques such as Language2Pose (Ahuja & Morency, 2019) to establish mappings between text and motion. With the introduction of generative models like VAE, researchers began to learn probabilistic distributions from text to motion; however, these methods still face challenges in modeling long-term motion sequences. Recently, Transformer-based architectures have substantially improved motion generation quality. For example, MotionDiffuse (Zhang et al., 2024a) utilizes diffusion models to achieve progressive denoising, making strides in generating natural and fluid motions; BAMM (Pinyoanuntapong et al., 2024) employs bidirectional autoregressive modeling to support more flexible sequence generation. Nonetheless, these methods continue to suffer from insufficient utilization of historical

information, particularly struggling to maintain long-term consistency in complex continuous motions.

State Space Models (SSM), due to their efficient handling of long sequences, have become a new research focus. Motion Mamba (Zhang et al., 2024d) enhances computational efficiency while preserving generation quality through hierarchical temporal scanning and bidirectional spatial modeling; InfiniMotion (Zhang et al., 2024c) further introduces memory-augmented mechanisms to cache historical motion features, strengthening sequence coherence. Although these methods perform well overall, they lack mechanisms to differentiate the importance of features across different temporal steps, making it difficult to adapt to the varying rhythms of motion.

## 2.2 DYNAMIC SEQUENCE MODELING

Advances in sequence modeling have opened new possibilities for motion generation. Early works like T2M-GPT (Zhang et al., 2023a) primarily rely on positional encoding to convey temporal information but struggle to effectively capture long-range dependencies. Recent research, such as MotionDiffuse (Zhang et al., 2024a), attempts to enhance temporal modeling through diffusion processes but still lacks robustness in handling changes in motion rhythm. Pose-guided Motion Diffusion (Cai et al., 2024) employs a pose memory bank for motion composition, while InfiniMotion (Zhang et al., 2024c) introduces memory modules to preserve historical states. Although these methods improve sequence modeling to some degree, they remain limited in dynamically integrating historical information and differentiating feature importance across temporal steps.

In summary, existing text-to-motion generation methods have achieved varying degrees of progress in generation quality, local motion control, and long-range dependency modeling. However, they commonly fall short in globally integrating historical information and dynamically assigning weights to temporal step features. These challenges are especially prominent when generating long, rhythmically diverse, and logically coherent motion sequences, resulting in deficiencies in coherence, naturalness, and semantic alignment. To address these issues, this paper proposes the Feedback-Augmented Decay Motion Model (FADM) framework, which more effectively exploits historical information and enhances the model's capability to weight temporal step features, thereby generating motion sequences that better align with textual semantics and exhibit higher coherence.

## 3 METHOD

### 3.1 OVERALL OVERVIEW

In recent years, text-driven human motion generation has witnessed rapid progress, with substantial gains in both motion quality and expressive diversity. Much of this progress stems from advances in deep generative modeling and sequence architectures. Nevertheless, autoregressive or masked Transformer backbones alone remain insufficient to achieve truly natural, coherent, and semantically faithful human motions. In particular, long-sequence generation requires a delicate balance between local smoothness and global semantic consistency, which places higher demands on historical context modeling and the adaptive weighting of temporal features.

Motivated by these challenges, we propose the Feedback-Augmented Decay Motion Model (FADM), which enhances the utilization of historical context and the dynamic weighting of temporal features through multi-scale feedback and decaying mechanisms.

Although the Masked Transformer (M-Transformer) demonstrates strong performance in generation tasks, its utilization of historical information in motion sequence modeling remains inadequate, particularly for capturing long-range dependencies. Moreover, its temporal step weighting is relatively static, making it difficult to adapt to the varying importance of motion features across different time steps during generation. These deficiencies directly affect the coherence of the generated sequences and their alignment with the input text.

To address these issues, FADM extends the M-Transformer with two core innovations, designed to simultaneously strengthen historical context modeling and dynamically allocate temporal importance, thereby enabling more natural and semantically consistent motion generation:

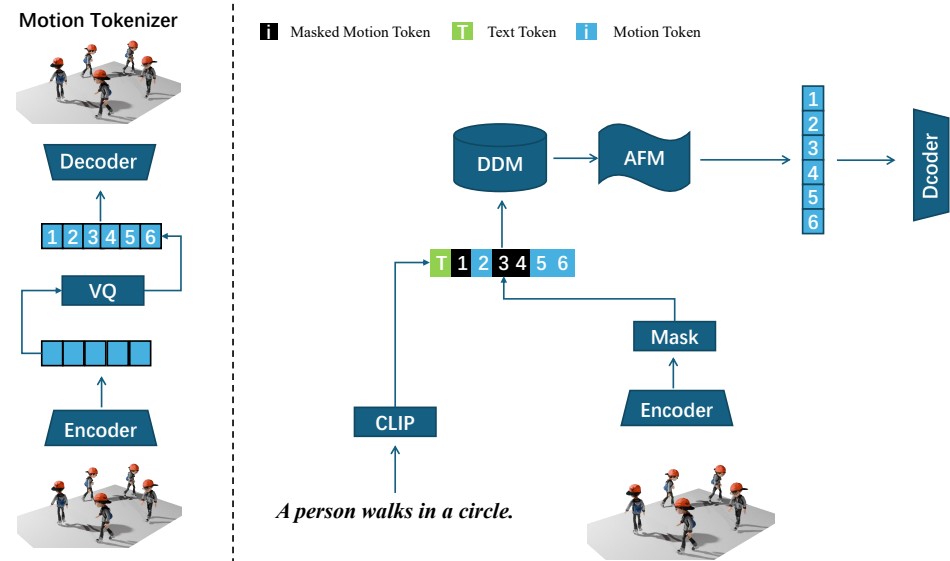

Figure 1: Overview of the Feedback-Augmented Decay Motion Model (FADM), highlighting the integration of the Multi-Scale Autoregressive Feedback Module (AFM) and the Dynamic Decay Module (DDM) to improve historical context modeling and dynamic temporal weighting in motion generation.

- Multi-Scale Autoregressive Feedback Module (AFM): Enhances the guidance from historically generated tokens on current decisions through multi-scale memory fusion and dynamic gating.
- Dynamic Decay Module (DDM): Applies exponential decay to temporal step weights in the input sequence to reduce the interference of early-step information on later generation stages.

In implementation, the model first extracts textual and initial motion features via an encoder, followed by processing through a masking module. Subsequently, the AFM module integrates historical information to strengthen contextual correlations during generation, while the DDM module dynamically adjusts temporal weights based on time-step characteristics. Together, these components enable the generation of high-quality motion sequences.

## 3.2 MULTI-SCALE AUTOREGRESSIVE FEEDBACK MECHANISM

Conventional masked models generate the current output by conditioning only on the present masked sequence and the provided text description. They do not explicitly leverage previously generated results, which can lead to reduced coherence in sequential motion synthesis. To address this limitation, we introduce the Multi-Scale Autoregressive Feedback Mechanism (AFM), illustrated in Figure 2. This mechanism explicitly integrates historical information with current conditions, and operates in three main steps as follows.

**Historical Feature Extraction.** The first step is to obtain a compact representation of prior motion context. During training, the initial $k$ time steps of the input sequence are treated as historical reference frames. We average their token features and project them into a latent space via a small network:

$$h_{\text{hist}} = FA\left(\frac{1}{k}\sum_{t=1}^{k}x_t\right) \tag{1}$$

Here, $x_t$ denotes the token feature at historical time step $t$, and $F_A$ is a two-layer linear network responsible for compressing these aggregated features. This allows the model to retain high-level motion semantics from the past.

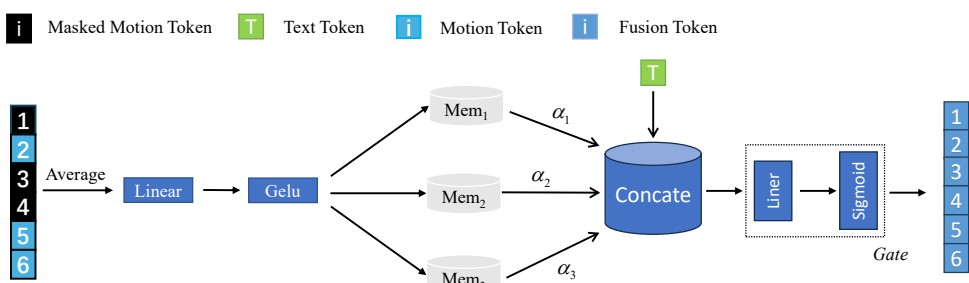

Figure 2: The structure of the Multi-Scale Autoregressive Feedback Mechanism (AFM), which enhances motion coherence by extracting historical features, applying multi-scale memory fusion, and dynamically gating feedback to balance past motion cues with current textual conditions.

**Multi-Scale Memory Fusion.** Realistic motions evolve at both global and local temporal scales. To capture these, we introduce three parallel memory modules, each applying a different level of smoothing to the historical representation:

$$m_i = \text{Mem}_i(h_{\text{hist}}), \quad i = 1, 2, 3 \tag{2}$$

where $\text{Mem}_i$ is a linear transformation layer, and the multi-scale memory is fused with a scale factor $\alpha_i = 0.5^i$ for weighted fusion:

$$h_{\text{mem}} = \sum_{i=1}^{3} \alpha_i \cdot m_i \tag{3}$$

This design enables the model to perceive both long-term trends and fine-grained dynamics. For example, when generating a "turning—kicking" sequence, the model can track the global rotation while preserving nuanced leg movements.

**Dynamic Feedback Gating.** Finally, to adaptively balance the influence between historical cues and the current textual condition, we apply a gating mechanism. The gate value is computed as:

$$g = \sigma \left( \text{Gate} \left( \text{concat}(h_{\text{mem}}, c) \right) \cdot s \right) \tag{4}$$

Here, $c$ is the text condition feature, Gate is a single-layer linear network, $\sigma$ is the Sigmoid function, and $s = \min \left( \frac{\text{len}}{20}, 1.0 \right)$ is the feedback intensity factor that increases with the sequence length. The final fused feature is:

$$c_{\text{updated}} = c \cdot (1 - g) + h_{\text{mem}} \cdot g \tag{5}$$

Through this mechanism, the model primarily relies on textual guidance at the beginning of generation, while progressively shifting attention toward maintaining historical coherence as the sequence lengthens.

### 3.3 DYNAMIC DECAY FACTOR

To address the issue of equal weighting assigned to all time steps in Transformer models, particularly the problem of early time steps excessively interfering with later generations, this paper proposes the design of a Dynamic Decay Module (DDM). This module introduces a weight distribution method that exponentially decays over time steps, effectively reducing the interference from historical time steps on the current generation.

It is important to note that the choice of an exponential decay function is not arbitrary. Its form aligns with the need to progressively decrease the interference from historical information as time progresses. Specifically, the exponential form in the equation expresses that, as the time steps increase, the decay factor rapidly diminishes, indicating that the influence of past information on the current time step follows a diminishing trend. This design draws inspiration from various natural phenomena, such as radioactive decay in physics, which adhere to similar exponential decay patterns. By employing this method, we ensure that the interference from early time steps on later

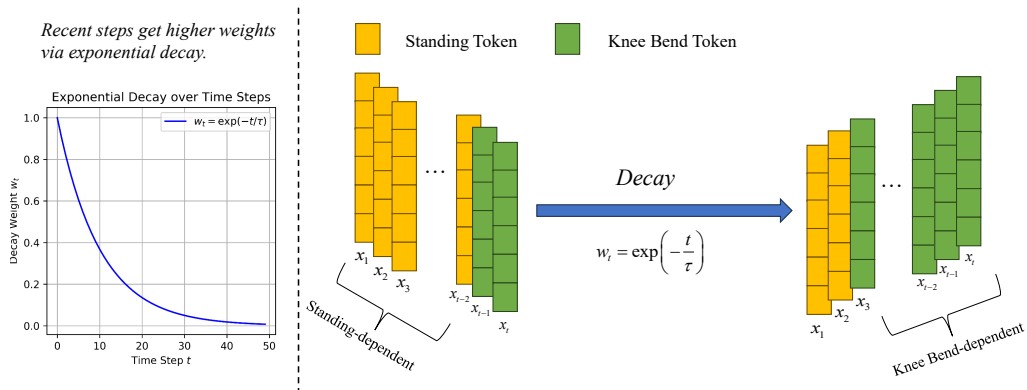

Figure 3: An illustration of the Dynamic Decay Module (DDM). The example shows how exponential decay weighting reduces the influence of early-step actions.

generations is effectively suppressed, thus enhancing the coherence and accuracy of the generated sequences.

As shown in Figure 3, the model introduces an exponential decay mechanism that allows it to focus more on recent action features during the generation process. For example, before the "jump" action, the model will give more attention to the features of the "kneel" action, rather than the earlier "standing" action features. This approach helps the model more accurately capture the key details of the current action, improving the coherence and naturalness of the generated results. To formalize this mechanism, we begin by introducing its specific implementation, starting from the decay weight design of the time steps:

**Time Step Decay Weight.** A decay weight is assigned to each time step $t$ of the input sequence:

$$w_t = \exp\left(-\frac{t}{\tau}\right) \tag{6}$$

where $\tau = 10$ is the decay factor, used to control the rate at which the weights decay over time steps. This design allows recent time steps to receive higher weights during the generation process, while the influence of earlier time steps gradually diminishes, ensuring that the model focuses more on the temporal information closely related to the current action during generation decisions.

**Weighted Input Sequence.** To emphasize the importance of different time steps in the generation process, the decay weight $w_t$ is applied to the input features:

$$\tilde{x}_t = x_t \cdot W_t \tag{7}$$

This weighting method effectively enhances the model's focus on recent temporal information that is highly relevant to the current action, while suppressing the detrimental interference of earlier time step features on the generation process.

### 3.4 LOSS FUNCTION

To ensure the logical coherence and semantic consistency of the generated sequence, this paper establishes a consistency constraint between historical and predicted features based on the loss design of MoMask, which is defined as follows:

$$\mathcal{L}_{\text{consist}} = \left\| h_{\text{hist}} - FA\left(\frac{1}{n}\sum_{t=1}^{n} \hat{x}_t\right) \right\|_2^2 \tag{8}$$

where $\hat{x}_t$ represents the predicted token features, and $h_{\text{hist}}$ denotes the historical features. This loss function is used to encourage the model to maintain alignment with historical trends during the generation process, thereby reducing semantic drift.

Table 1: Quantitative Evaluation Results on HumanML3D Dataset

| Datasets | Methods | R Precision ↑ | | | FID ↓ | MM-Dist ↓ | MultiModality ↑ |
|---|---|---|---|---|---|---|---|
| | | Top 1 | Top 2 | Top 3 | | | |
| | Ground Truth | $0.511^{\pm.003}$ | $0.703^{\pm.003}$ | $0.797^{\pm.002}$ | $0.002^{\pm.000}$ | $2.974^{\pm.008}$ | - |
| Human ML3D | TM2T (Gong et al., 2023) | $0.424^{\pm.003}$ | $0.618^{\pm.003}$ | $0.729^{\pm.002}$ | $1.501^{\pm.017}$ | $3.467^{\pm.011}$ | $\underline{2.424}^{\pm.093}$ |
| | T2M (Guo et al., 2022) | $0.455^{\pm.003}$ | $0.636^{\pm.003}$ | $0.736^{\pm.002}$ | $1.087^{\pm.021}$ | $3.347^{\pm.008}$ | $2.219^{\pm.074}$ |
| | MDM (Tevet et al., 2022) | $0.320^{\pm.005}$ | $0.498^{\pm.004}$ | $0.611^{\pm.007}$ | $0.544^{\pm.044}$ | $5.566^{\pm.027}$ | $\mathbf{2.799}^{\pm.072}$ |
| | MLD (Chen et al., 2023) | $0.481^{\pm.003}$ | $0.673^{\pm.003}$ | $0.772^{\pm.002}$ | $0.473^{\pm.013}$ | $3.196^{\pm.010}$ | $2.413^{\pm.079}$ |
| | MotionDiffuse (Zhang et al., 2024a) | $0.491^{\pm.001}$ | $0.681^{\pm.001}$ | $0.782^{\pm.001}$ | $0.630^{\pm.001}$ | $3.113^{\pm.001}$ | $1.553^{\pm.042}$ |
| | ReMoDiffuse (Zhang et al., 2023b) | $0.510^{\pm.005}$ | $0.698^{\pm.006}$ | $0.795^{\pm.004}$ | $0.103^{\pm.004}$ | $2.974^{\pm.016}$ | $1.795^{\pm.043}$ |
| | T2M-GPT (Zhang et al., 2023a) | $0.492^{\pm.003}$ | $0.679^{\pm.002}$ | $0.775^{\pm.002}$ | $0.141^{\pm.005}$ | $3.121^{\pm.009}$ | $1.831^{\pm.048}$ |
| | MotionGPT (Jiang et al., 2023) | $0.492^{\pm.003}$ | $0.681^{\pm.003}$ | $0.778^{\pm.002}$ | $0.232^{\pm.008}$ | $3.096^{\pm.008}$ | $2.008^{\pm.084}$ |
| | ParCo (Zou et al., 2024) | $0.515^{\pm.003}$ | $0.706^{\pm.003}$ | $0.801^{\pm.002}$ | $0.109^{\pm.005}$ | $\mathbf{2.927}^{\pm.008}$ | $1.382^{\pm.060}$ |
| | Motion Mamba (Zhang et al., 2024d) | $0.502^{\pm.003}$ | $0.693^{\pm.002}$ | $0.792^{\pm.002}$ | $0.281^{\pm.009}$ | $3.060^{\pm.058}$ | $2.294^{\pm.058}$ |
| | MoMask (Guo et al., 2024) | $\underline{0.521}^{\pm.002}$ | $\underline{0.713}^{\pm.002}$ | $\underline{0.807}^{\pm.002}$ | $\underline{0.045}^{\pm.002}$ | $2.958^{\pm.008}$ | $1.241^{\pm.040}$ |
| | **Ours** | $\mathbf{0.527}^{\pm.003}$ | $\mathbf{0.716}^{\pm.002}$ | $\mathbf{0.810}^{\pm.001}$ | $\mathbf{0.035}^{\pm.003}$ | $\underline{2.936}^{\pm.007}$ | $1.283^{\pm.044}$ |

Table 2: Quantitative Evaluation Results on KIT-ML Dataset

| Datasets | Methods | R Precision ↑ | | | FID ↓ | MM-Dist ↓ | MultiModality ↑ |
|---|---|---|---|---|---|---|---|
| | | Top 1 | Top 2 | Top 3 | | | |
| | Ground Truth | $0.424^{\pm.005}$ | $0.649^{\pm.006}$ | $0.779^{\pm.006}$ | $0.031^{\pm.004}$ | $2.788^{\pm.012}$ | - |
| KIT-ML | TM2T (Gong et al., 2023) | $0.280^{\pm.005}$ | $0.463^{\pm.006}$ | $0.587^{\pm.005}$ | $3.599^{\pm.153}$ | $4.591^{\pm.026}$ | $\mathbf{3.292}^{\pm.081}$ |
| | T2M (Guo et al., 2022) | $0.361^{\pm.005}$ | $0.559^{\pm.007}$ | $0.681^{\pm.007}$ | $3.022^{\pm.107}$ | $3.488^{\pm.028}$ | $2.052^{\pm.107}$ |
| | MDM (Tevet et al., 2022) | $0.164^{\pm.004}$ | $0.291^{\pm.004}$ | $0.396^{\pm.004}$ | $0.497^{\pm.021}$ | $9.191^{\pm.022}$ | $1.907^{\pm.214}$ |
| | MLD (Chen et al., 2023) | $0.390^{\pm.008}$ | $0.609^{\pm.008}$ | $0.734^{\pm.007}$ | $0.404^{\pm.027}$ | $3.204^{\pm.027}$ | $\underline{2.192}^{\pm.071}$ |
| | MotionDiffuse (Zhang et al., 2024a) | $0.417^{\pm.004}$ | $0.621^{\pm.004}$ | $0.739^{\pm.004}$ | $1.954^{\pm.062}$ | $2.958^{\pm.005}$ | $0.730^{\pm.013}$ |
| | ReMoDiffuse (Zhang et al., 2023b) | $0.427^{\pm.014}$ | $0.641^{\pm.004}$ | $0.765^{\pm.055}$ | $\underline{0.155}^{\pm.006}$ | $2.814^{\pm.012}$ | $1.239^{\pm.028}$ |
| | ParCo (Zou et al., 2024) | $0.430^{\pm.004}$ | $0.649^{\pm.007}$ | $0.772^{\pm.006}$ | $0.453^{\pm.027}$ | $2.820^{\pm.028}$ | $1.245^{\pm.022}$ |
| | Motion Mamba (Zhang et al., 2024d) | $0.419^{\pm.006}$ | $0.645^{\pm.005}$ | $0.765^{\pm.006}$ | $0.307^{\pm.041}$ | $3.021^{\pm.025}$ | $1.678^{\pm.064}$ |
| | MoMask (Guo et al., 2024) | $\underline{0.433}^{\pm.007}$ | $\underline{0.656}^{\pm.005}$ | $\underline{0.781}^{\pm.005}$ | $0.204^{\pm.011}$ | $\underline{2.779}^{\pm.022}$ | $1.131^{\pm.043}$ |
| | **Ours** | $\mathbf{0.437}^{\pm.003}$ | $\mathbf{0.659}^{\pm.003}$ | $\mathbf{0.783}^{\pm.002}$ | $\mathbf{0.141}^{\pm.003}$ | $\mathbf{2.761}^{\pm.008}$ | $1.157^{\pm.062}$ |

Table 3: Ablation Study on M-Transformer

| Methods | R Precision ↑ | | | FID ↓ | MM-Dist ↓ | MultiModality ↑ |
|---|---|---|---|---|---|---|
| | Top 1 | Top 2 | Top 3 | | | |
| w/o DDM | $0.520^{\pm.003}$ | $0.713^{\pm.002}$ | $0.806^{\pm.002}$ | $0.037^{\pm.002}$ | $2.939^{\pm.008}$ | $1.293^{\pm.050}$ |
| w/o AFM | $0.515^{\pm.003}$ | $0.710^{\pm.002}$ | $0.804^{\pm.002}$ | $0.045^{\pm.002}$ | $2.979^{\pm.006}$ | $\mathbf{1.350}^{\pm.043}$ |
| **Ours** | $\mathbf{0.527}^{\pm.003}$ | $\mathbf{0.716}^{\pm.002}$ | $\mathbf{0.810}^{\pm.001}$ | $\mathbf{0.035}^{\pm.003}$ | $\mathbf{2.936}^{\pm.007}$ | $1.283^{\pm.044}$ |

The total loss function of the model consists of three components:

$$\mathcal{L}_{\text{total}} = \mathcal{L}_{\text{mask}} + \mathcal{L}_{\text{res}} + \mathcal{L}_{\text{consist}} \tag{9}$$

where $\mathcal{L}_{\text{mask}}$ is the mask prediction loss, which is the negative log-likelihood of predicting the masked tokens; $\mathcal{L}_{\text{res}}$ is the residual layer prediction loss; and $\mathcal{L}_{\text{consist}}$ is the semantic consistency loss. Through the joint optimization of these three components, the model not only maintains its baseline performance but also more effectively utilizes historical information, improving the coherence of the generated sequence and its alignment with the text semantics.

## 4 EXPERIMENTS

**Dataset.** This study conducted experiments using two commonly used public text-action datasets: the HumanML3D dataset and the KIT-ML dataset. The HumanML3D dataset contains 14,616 action sequences and 44,970 text descriptions, covering a variety of human activities such as sports, dance, and acrobatics. The data is sourced from the AMASS (Mahmood et al., 2019) and Human-Act12 (Guo et al., 2020) datasets, and each action sequence has been standardized. The KIT-ML dataset is smaller in scale, containing 3,911 action sequences and 6,278 text descriptions. Both datasets are divided into training, validation, and test sets in the ratio of 80%, 15%, and 5%, respectively.

**Evaluation Metrics.** Based on the T2M evaluation framework, this paper extracts features using the same pre-trained text-action encoder and computes the following multimodal metrics: (1) **Frechet Inception Distance (FID)** quantifies the overall generation quality by comparing the distribution differences between the generated actions and the real actions in the high-level semantic space; (2) **R-Precision** and **Multimodal Distance (MM-Dist)** assess the semantic consistency between the generated sequences and the input text in terms of retrieval accuracy and feature distance, respectively; (3) **Multimodality**, by repeatedly sampling the same text input, calculates the average Euclidean distance between generated action pairs to measure the model's response diversity to potential textual ambiguities.

## 4.1 IMPLEMENTATION DETAILS

The proposed framework is implemented using PyTorch and trained on a single NVIDIA GeForce RTX 4090 GPU. During training on the HumanML3D dataset, the batch size is set to 64, while it is set to 32 for the KIT-ML dataset. The learning rate for all models is linearly warmed up over the first 2000 iterations to reach $2 \times 10^{-4}$, followed by a learning rate decay strategy where the learning rate is reduced by a factor of 0.1 after 50,000 iterations. The training process spans a total of 2000 epochs. The latent dimension of the Transformer is set to 384, and the dropout rate is configured to 0.2. Model checkpoints are saved every 500 iterations to facilitate model recovery and performance evaluation.

## 4.2 EXPERIMENTAL RESULTS AND ANALYSIS

To assess the effectiveness of the proposed method, we compare it with several state-of-the-art models, including VAE-based, diffusion-based, and autoregressive models. All experiments are conducted under standard protocols, with each experiment repeated 20 times. The results are reported as mean values along with the 95% confidence intervals.

### 4.2.1 QUANTITATIVE EVALUATION

As shown in Table 1, the experimental results demonstrate that FADM outperforms existing mainstream methods across the board. Compared to MoMask, FADM significantly reduces the FID score from 0.045 to 0.035, representing a relative improvement of 22.2%, which validates that the generated motions align more closely with real data in terms of distribution. In terms of semantic understanding, the model achieves a Top-1 accuracy of 0.527, reflecting a 64.7% improvement over the baseline, while the Top-3 accuracy reaches 0.810. These substantial gains confirm the model's ability to capture fine-grained semantics from text. Notably, FADM maintains a competitive advantage in the Multimodality metric while simultaneously reducing the MM-Dist score. This result indicates that the model effectively controls motion distortion caused by semantic deviation, without sacrificing diversity in response to textual ambiguity.

Table 2 demonstrates the experimental results, further validating the generalization capability of FADM. On the KIT-ML dataset, the model demonstrates three core advantages: In terms of generation quality, the FID score is reduced by 30.9% compared to MoMask, and a significant improvement of 54.1% is achieved compared to Mamba, proving that the generated motion quality closely approximates the real data distribution. Regarding semantic alignment, the Top-3 accuracy of 0.783 represents a 97.8% improvement over MDM, highlighting a higher degree of text-action alignment. In terms of multimodal expression capability, our method shows a slight gap compared to Motion Mamba, but when combined with the superior MM-Dist metric, FADM achieves a better balance between quality and diversity through precise semantic control.

### 4.2.2 ABLATION STUDY

To verify the necessity of the multi-scale autoregressive feedback mechanism and dynamic decay factor, we designed an ablation study comparing the performance differences between the model with "removed dynamic decay factor," "removed multi-scale autoregressive feedback mechanism," and the complete model. The results are shown in Table 3. The model without the dynamic decay factor, which removes the time step weight exponential decay mechanism, showed an increase in FID (from 0.035 to 0.037) and a decrease in Top accuracy due to early information interference.

This confirms that the dynamic decay factor improves generation quality and semantic alignment by emphasizing recent information. The model without the multi-scale autoregressive feedback mechanism failed to utilize the semantic features of historical generated tokens, with FID increasing to 0.045 and a significant drop in Top accuracy, highlighting the critical role of the multi-scale autoregressive feedback mechanism in maintaining sequence action coherence through the integration of global historical context. The complete model outperformed in all metrics, demonstrating that both modules work collaboratively through "precise reuse of historical information and dynamic adjustment of time step weights" to solve core issues in sequence generation, validating the rationale of the framework design and emphasizing the indispensability of these two innovative modules in improving generation quality and enhancing text-motion alignment.

## 5 CONCLUSION

The proposed FADM framework, comprising the Multi-Scale Autoregressive Feedback Module (AFM) and the Dynamic Decay Module (DDM), effectively addresses two core challenges in text-to-motion generation: insufficient utilization of historical information and imbalanced temporal step weighting. By leveraging multi-scale memory fusion, dynamic gating techniques, and decay factors, the model systematically improves the logical continuity and dynamic adaptability of generated motions. Experimental results on the HumanML3D dataset demonstrate that FADM not only achieves a substantial 22.2% reduction in FID but also attains a Top-1 accuracy improvement of 64.7%. These outcomes validate that the weighted feature summation combined with an exponential decay mechanism can simultaneously enhance sequence coherence and semantic alignment.

Notably, the technical approach proposed in this study has broad applicability. On one hand, the multi-scale autoregressive feedback mechanism provides a scalable solution for modeling long-term dependencies via multi-scale memory fusion, making it suitable for tasks requiring sustained long-range consistency, such as video generation and dialogue systems. On the other hand, the dynamic decay factor's differentiated weighting of temporal steps offers a novel perspective for generation tasks that demand fine-grained rhythm control, including music synthesis and sign language recognition. Future work will further explore the universal applicability of memory-augmented mechanisms in cross-modal generation and optimize computational efficiency in real-time generation scenarios, thereby advancing human-computer interaction experiences in virtual reality and the metaverse.

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

## A  APPENDIX

### A.1  THE USE OF LLMS

In this work, large language models were used only to assist in polishing the writing. All ideas, analyses, and conclusions are the authors' own.

