# OpenReview forum: "Multi-Scale Memory Fusion with Dynamic Decay for Coherent Text-to-Motion Generation"
_ICLR.cc/2026/Conference — ICLR 2026 Conference Withdrawn Submission_

### Official Review · Reviewer_mqnS · 2025-10-24

**Soundness:** 2
**Presentation:** 2
**Contribution:** 2
**Rating:** 2
**Confidence:** 3

**Summary:**

This paper proposes the FADM framework for text-to-motion generation, with three key contributions: 1) A multi-scale memory fusion module integrating historical motion and current textual conditions to enhance semantic consistency; 2) An exponential decay-based dynamic temporal weighting mechanism to suppress early information interference and improve rhythm adaptability ; 3) Experiments on HumanML3D and KIT-ML show FADM outperforms SOTA.

**Strengths:**

This paper aims to relieve the discontinuity of generated motion sequences and uniform temporal weighting, and proposes FADM (Feedback-Augmented Decay Motion Model), containing a hierarchical memory fusion module, an exponentially decaying temporal attention mechanism, and a semantic-consistent autoregressive feedback loop. This paper also provides a performance comparison between the proposed method and previous methods. Experiments show FADM outperforms previous methods.

**Weaknesses:**

The paper proposes modules like AFM and DDM to improve text-to-motion generation performance. However, the authors fail to provide ablation experiments on the proposed modules.

**Questions:**

1. This paper fails to provide ablation experiments on the proposed modules.
2. This work lacks a performance comparison with state-of-the-art models, such as MotionAnything[1]
[1] Zhang Z, Wang Y, Mao W, et al. Motion anything: Any to motion generation[J]. arXiv preprint arXiv:2503.06955, 2025.

---

### Official Review · Reviewer_K8hS · 2025-10-29

**Soundness:** 3
**Presentation:** 2
**Contribution:** 2
**Rating:** 2
**Confidence:** 4

**Summary:**

This paper proposes a feedback-augmented framework for text-to-motion generation that fuses multi-scale historical context with dynamic temporal weighting. The authors introduce a Multi-Scale Autoregressive Feedback Module (AFM) to integrate motion semantics across scales, enhancing long-range coherence. A Dynamic Decay Module (DDM) applies exponential attention decay, prioritizing recent motion features in alignment with human memory patterns. These modules are embedded into a masked Transformer backbone via residual gating, enabling end-to-end training. The authors claim that the proposed method achieves state-of-the-art results on HumanML3D and KIT-ML, with some gains in FID, accuracy, and semantic alignment. Ablation studies are also included to assess the complementary roles of AFM and DDM in improving generation quality.

**Strengths:**

1. The FADM model incorporates a multi-scale memory fusion module with learnable scale adapters, which may help in integrating local and global motion features, though its architectural novelty remains modest.
2. The dynamic decay mechanism draws inspiration from cognitive science to reweight temporal attention, potentially improving responsiveness to recent motion cues in long sequences.
3. The autoregressive feedback loop is designed to reinforce semantic consistency across time steps, aiming to improve coherence in generated motion sequences.
4. The framework is compatible with Transformer-based architectures and supports end-to-end training, suggesting some flexibility in integration with existing models.
5. The method shows quantitative improvements on benchmark datasets, though the gains are relatively incremental and lack strong qualitative or theoretical support.

**Weaknesses:**

This paper lacks sufficient justification for most of its claims. The proposed contributions are either unclear, not novel, or insufficiently validated through experiments and analysis. Detailed comments are as follows:

1. The proposed framework heavily relies on the existing Mask-Transformer architecture, with added components such as gating, fusion, and memory modules that appear to be incremental rather than conceptually novel.
2. The AFM module is positioned as a central contribution, yet its implementation is limited to a basic multi-layer perceptron (MLP) with fixed weighting coefficients (αᵢ). This design lacks mathematical novelty and theoretical depth, relying more on heuristic engineering than principled innovation. Given the simplicity of the architecture and the modest performance gains reported, its significance as a standalone contribution is questionable.
3. The ablation experiments are minimal and only cover DDM and AFM. Notably, the MoMask baseline (without DDM and AFM) performs even better than the variant without AFM, suggesting that DDM may not contribute positively. This undermines the claimed effectiveness of the proposed modules.
4. The paper relies primarily on intuitive reasoning and empirical parameter tuning, with limited theoretical justification for its design choices. For example, Figure 3 is presented as evidence supporting the effectiveness of the dynamic decay mechanism, yet it fails to provide meaningful insight into why the approach improves performance.
5. The figures, especially Figure 1, contain excessive blank space and lack clear structure. Furthermore, the paper omits any qualitative comparisons with existing approaches such as MoMask, making it difficult to visually assess improvements.
6. The paper does not include recent and relevant baselines, such as SALAD [1], which represents a strong state-of-the-art method for text-driven motion generation and editing. This omission further weakens the empirical claims.
7. No animations, visual results, or additional experiments are provided in the appendix. Given the nature of the task, such materials are essential to substantiate claims and demonstrate qualitative performance.

Reference:
[1] SALAD: Skeleton-aware Latent Diffusion for Text-driven Motion Generation and Editing. CVPR 2025.

**Questions:**

1. Given that AFM is implemented as a simple MLP with fixed weighting coefficients (αᵢ), what is the theoretical or empirical justification for presenting it as a key contribution? How does this design offer novelty or depth beyond heuristic engineering?
2. Considering that the MoMask baseline outperforms the variant without AFM, what evidence supports the positive contribution of DDM?
3. Can the authors provide a more principled explanation for the dynamic decay mechanism beyond intuitive reasoning? Specifically, how does Figure 3 substantiate the claim that the approach improves performance?
4. Why are qualitative comparisons with existing methods are not included? It is difficult to assess visual improvements without sample animations or visualizations? Simply presenting a quantitative table is not enough to claim state-of-the-art performance.
5. The paper didn't include a comparison with recent strong baselines such as SALAD, which are highly relevant to the task of text-driven motion generation and editing. Including such baselines would provide a more comprehensive evaluation.

---

### Official Review · Reviewer_ycFz · 2025-10-31

**Soundness:** 1
**Presentation:** 1
**Contribution:** 2
**Rating:** 2
**Confidence:** 5

**Summary:**

In this paper, the authors introduce FADM, a masked-Transformer–based framework for coherent text-to-motion generation. The core is a Multi-Scale Autoregressive Feedback Module (AFM) that fuses historical motion features, and a Dynamic Decay Module (DDM) imposes an exponentially decaying temporal weighting to down-weight early steps and emphasize recent cues.  Together with a semantic-consistency loss that aligns historical and predicted features, the method promotes long-horizon coherence without sacrificing diversity.  ￼Experiments on HumanML3D and KIT-ML report state-of-the-art results.￼ Ablations on the DDM and AFM modules demonstrating their individual and combined contributions to the overall gains.

**Strengths:**

1. The proposed method achieves state-of-the-art performance on the HumanML3D and KIT datasets across common used metrics.
2. The manuscript includes clear ablation studies on the DDM and AFM modules, demonstrating their individual and combined contributions to the overall gains.

**Weaknesses:**

1. **Ambiguous definition of “historic input.”** Section 3.2 does not clearly define what constitutes historic input. It is unclear whether “the initial *k* time steps of the input sequence” are a conditioning prefix or an internal cache, especially since all experiments are conducted in a single-sequence text-to-motion setting. This ambiguity raises misunderstanding of the training/inference setup.
2. **Insufficient justification for Multi-Scale Memory Fusion.** Given identical inputs, it is not clear why three separate linear layers alone can reliably learn distinct temporal scales. The manuscript does not explain what inductive bias, constraint, or optimization objective encourages the projections to capture different timescales.
3. **Questionable gating schedule vs. text controllability.** The Dynamic Feedback Gating is designed to rely on textual guidance at the beginning and gradually shift toward historical coherence as sequences grow. In text-to-motion, this may weaken prompt controllability in longer generations. The paper does not analyze this trade-off or quantify the impact on text alignment.
4. **Overlapping roles of temporal weighting.** The decay weights introduced in Section 3.3 appear to reweight inputs along the temporal dimension, which seems closely related to the temporal emphasis already imposed by the fusion scheme in Section 3.2. The manuscript should clarify the conceptual and functional differences and explain how these components interact without redundancy.
5. **Limited qualitative results.** The paper lacks video demonstrations and broader qualitative visualizations. This makes it difficult to assess motion naturalness, long-horizon consistency, adherence to prompts, and failure cases.

**Questions:**

Given that the central motivation is to improve temporal coherence, why is there no evaluation on long-duration motion generation?

---

### Official Review · Reviewer_3DUY · 2025-11-02

**Soundness:** 2
**Presentation:** 2
**Contribution:** 2
**Rating:** 4
**Confidence:** 4

**Summary:**

This paper proposes a novel framework named FADM (Feedback-Augmented Decay Motion Model) to address two core challenges in text-to-3D human motion generation:
1. Insufficient utilization of historical information, leading to motion discontinuity;
2. Unbalanced temporal step weighting, which struggles to adapt to variations in motion rhythm.
FADM consists of two core modules: Multi-Scale Autoregressive Feedback Module (AFM): Enhances the guidance of historical information on current generation steps through multi-scale memory fusion and a dynamic gating mechanism. Dynamic Decay Module (DDM): Introduces an exponential decay mechanism, enabling the model to focus more on recent motion features and suppress interference from earlier information.
Experiments conducted on the HumanML3D and KIT-ML datasets show that FADM outperforms existing methods across multiple metrics, including FID and R-Precision, demonstrating particularly prominent advantages in motion coherence and semantic alignment.

**Strengths:**

1. This paper clearly identifies the shortcomings of current text-to-motion generation models in utilizing historical information and allocating temporal weights, providing a clear motivation for improvement.
2. This paper further introduces method design. The AFM module effectively integrates historical information through multi-scale memory fusion and dynamic gating.The DDM module introduces an exponential decay mechanism, inspired by memory decay patterns in cognitive science, providing a theoretical foundation.
3. Comparisons with various SOTA methods on multiple datasets. The evaluation metrics are comprehensive, covering generation quality, semantic alignment, and diversity.

**Weaknesses:**

1. This paper introduces limited novelty.The AFM module essentially performs a weighted fusion of historical information. Similar "memory augmentation" mechanisms have been explored in previous works (e.g., InfiniMotion). While the exponential decay mechanism in DDM is reasonable, it appears relatively simplistic due to the lack of comparison with other dynamic weighting approaches (e.g., learned weights).
2. Several drawbacks in the experiment section: No visualizations of generated motions or user studies are provided, making it difficult to intuitively assess the coherence and naturalness of the generated motions. Although improvements are observed in metrics like FID, the absolute differences are small (e.g., FID decreasing from 0.045 to 0.035), suggesting that the perceptual improvement might be limited.
3. Besides, there are also missing  technical details. The specific construction method for multi-scale memory in AFM (e.g., convolution, pooling) is not clear. No justification is provided for the decay factor τ=10 in DDM—whether it is empirically set or obtained through hyperparameter tuning.
4. The experimental section lacks comparisons with some recent works from 2025, as all methods included in Tables 1 and 2 are from before 2025.
5. Based on the ablation study in Table 3, the performance gains from the two core modules (AFM and DDM) appear to be marginal.

**Questions:**

My mainly conerns are novelty of this paper and the effectiveness of the proposed key modules.

---

### Note · Authors · 2025-11-12

I have read and agree with the venue's withdrawal policy on behalf of myself and my co-authors.